# Antitumor Peptide-Based Vaccine in the Limelight

**DOI:** 10.3390/vaccines10010070

**Published:** 2022-01-03

**Authors:** Takumi Kumai, Hidekiyo Yamaki, Michihisa Kono, Ryusuke Hayashi, Risa Wakisaka, Hiroki Komatsuda

**Affiliations:** 1Department of Innovative Head & Neck Cancer Research and Treatment, Asahikawa Medical University, Asahikawa 078-8510, Japan; 2Department of Otolaryngology-Head and Neck Surgery, Asahikawa Medical University, Midorigaoka-Higashi 2-1-1-1, Asahikawa 078-8510, Japan; hidekiyo@asahikawa-med.ac.jp (H.Y.); mkono@asahikawa-med.ac.jp (M.K.); ryuhayashi@asahikawa-med.ac.jp (R.H.); r-wakisaka@asahikawa-med.ac.jp (R.W.); komatsuda@asahikawa-med.ac.jp (H.K.)

**Keywords:** immunotherapy, peptide vaccine, tumor-associated antigen, adjuvant

## Abstract

The success of the immune checkpoint blockade has provided a proof of concept that immune cells are capable of attacking tumors in the clinic. However, clinical benefit is only observed in less than 20% of the patients due to the non-specific activation of immune cells by the immune checkpoint blockade. Developing tumor-specific immune responses is a challenging task that can be achieved by targeting tumor antigens to generate tumor-specific T-cell responses. The recent advancements in peptide-based immunotherapy have encouraged clinicians and patients who are struggling with cancer that is otherwise non-treatable with current therapeutics. By selecting appropriate epitopes from tumor antigens with suitable adjuvants, peptides can elicit robust antitumor responses in both mice and humans. Although recent experimental data and clinical trials suggest the potency of tumor reduction by peptide-based vaccines, earlier clinical trials based on the inadequate hypothesis have misled that peptide vaccines are not efficient in eliminating tumor cells. In this review, we highlighted the recent evidence that supports the rationale of peptide-based antitumor vaccines. We also discussed the strategies to select the optimal epitope for vaccines and the mechanism of how adjuvants increase the efficacy of this promising approach to treat cancer.

## 1. Introduction

The recent in-depth understanding of basic immunology has resulted in the translation of cancer immunotherapy from the bench to clinical practice. Immune checkpoint inhibitors (ICIs) have been applied in a variety of cancer types and have shown clinical benefits in otherwise untreatable patients with advanced cancer. Since ICIs are considered as the fourth step in cancer therapy after surgery, radiation, and chemotherapy, the responders to ICIs are limited to approximately 20% [1]. A limitation of this treatment is that ICIs alone cannot selectively activate cancer-specific T-cells. Autoimmune T-cells activated by ICIs cause immune-related adverse effects. Irrelevant T-cells that react to pathogenic microorganisms or non-self-antigens possess high-affinity T-cell receptors (TCRs) and compete with tumor-reactive T-cells. Since self-reactive T-cells, including tumor-reactive T-cells, usually have low-affinity TCRs, the selective activation of these cells is necessary to achieve antitumor activity by overwhelming irrelevant T-cells for antigen-presenting cells (APCs) and cytokine competition [2]. In order to stimulate tumor-reactive T-cells, TCR stimulation via tumor-derived antigens (signal 1) with co-stimulatory molecules (signal 2) and cytokines (signal 3) is necessary.

T-cells recognize the target antigens via an epitope, a key amino acid sequence that binds to the major histocompatibility complex (MHC), followed by TCR stimulation. Epitopes from intracellular antigens are presented on MHC class I through the proteasome in APCs (or tumor) to stimulate CD8 T-cells (CTLs). Recently, the upregulation of immunoproteasome has been reported as a positive biomarker to identify ICI-responders [3]. Unlike MHC class I, the epitope peptide presented on MHC class II is processed from extracellular antigens through endosomes to stimulate CD4 T-cells (HTLs). In addition to these classical pathways, dendritic cells (DCs), representative of professional APCs, have the ability to present exogenous antigens through phagosomes to CTLs via MHC class I (cross presentation) [4]. The lengths of peptides that bind to MHC class I and class II are 8–10 mer and 12–16 mer, respectively. The epitope peptides from tumor-associated antigens (TAAs) can be identified using an in silico analysis [5]. The immunogenicity of the predicted epitope is confirmed by establishing peptide-specific T-cell clones with synthetic peptides [6]. In theory, the administration of these peptides could be used as antitumor peptide vaccines that induce endogenous tumor-specific T-cell responses. The peptide vaccines have advantages over other tumor immunotherapies because of their tumor-specificity, simplicity, and cost-effectiveness; however, there is still room for the improvement in the application of this therapy as a standard option in cancer therapy. In this review, we will discuss the strategies to select appropriate peptides and immune adjuvants for antitumor immunity and the current progress in peptide-based tumor immunotherapy.

## 2. Selecting Optimal Peptides for Cancer Peptide Vaccines

Based on anchor residues and other frequent amino acids, CD4 and CD8 T-cell epitopes can be predicted using a computer algorithm [7]. Several algorithms are available online to predict the epitopes from the whole amino acid sequence of TAAs (e.g., http://tools.iedb.org/mhci/, http://www.syfpeithi.de, accessed on 31 December 2021). Although the history of these predicting methods starts from the early 2000s [8], the quality of prediction has improved. Currently, NetMHCpan-4.1 and NetMHCIIpan-4.0 are considered as reliable methods by combining the data from MHC-binding affinity and mass spectrometry-eluted ligands to predict CD8 and CD4 epitopes, respectively [9]. Although these algorithms are useful for predicting the epitope sequences, Chaves et al. reported that there are huge disparities in the predicted scores among algorithms [10]. They found a high degree of false-negative and false-positive predictions in each algorithm by using a mouse influenza model. The potential of false-negative results using these algorithms has also been suggested in a melanoma model [11]. Nevertheless, these algorithms are efficient tools to estimate possible epitopes that could elicit antitumor T-cell responses when combined with biological assays, which confirms that the corresponding epitope peptides can actually induce T-cell responses [12,13].

### 2.1. Selecting Tumor Antigens as a Source of Vaccines

Although the origin of the tumor is self-tissue, cancer and hematologic malignancies can be targets of peptide vaccines. As tumors utilize signaling pathways in an unregulated manner, these signaling proteins, including TAAs, are overexpressed in tumors compared to normal tissues. Since most of self-antigen-reactive T-cells are deleted in the thymus by negative selection to prevent autoimmune diseases, low-affinity self-antigen-reactive T-cells that can only react to tumor cells with high TAAs expression exist in cancer patients [14]. Existing in healthy donors without any clinical symptoms, these T-cells that escape negative selection might react only with tumors that express a high amount of antigens but not with normal tissues, which express relatively low amounts of TAAs, including the cancer–testis antigens and cancer stem-like antigens [15,16]. Recently, mutation-derived peptides (neoantigens) have been immunologically regarded as non-self-antigens (tumor-specific antigens (TSAs)) [17].

Tumors express an abundance of cell signaling proteins such as the epidermal growth factor (EGFR), which is expressed in healthy tissues to a relatively low extent [18]. As T-cells are not deleted when a self-antigen is expressed at a low concentration [19], these TAAs-reactive T-cells may recognize tumors, but not healthy tissues. Previously, we generated EGFR-specific HTLs from healthy donors using algorithm-predicted epitope peptides [14]. We confirmed the reactivity and cytotoxicity of EGFR-specific HTLs against tumor cell lines and the presence of EGFR-specific HTLs in head and neck cancer patients, suggesting that self-antigen-reactive T-cells exist and can recognize tumors. Notably, none of the donors and patients recruited in the study had autoimmune diseases related to the EGFR. Using the EGFR as a TAA model, the frequency of EGFR-specific CTLs was significantly correlated with the EGFR expression in head and neck cancer, strengthening the idea of targeting self-antigens as a TAA [20]. Therefore, even if target proteins are expressed in healthy tissues to some extent, the aberrant expression of these proteins in cancer can induce antitumor T-cells that only react to tumors.

Among TAAs, the expression of cancer-testis antigens such as MAGE and NY-ESO-1 is restricted in tumors and testes, but not in adult somatic tissues, indicating that this type of antigen is a favorable target for tumor immunotherapy [21]. Because CTL responses against cancer–testis antigens naturally occur in melanoma patients [22], the immune system can spontaneously recognize cancer–testis antigens as an immunogen. Cancer stem cells (CSCs) frequently express cancer–testis antigens. CSCs play a role in tumor initiation and chemotherapy resistance within a heterogeneous cancer population [23]. Targeting CSCs by cancer–testis antigen-based immunotherapy has shown superior antitumor effects than targeting survivin, a well-described noncancer testis TAA in a mouse model [16].

The most evident answer for unregulated tumor proliferation is the gain-of-function mutation of the signaling pathway [24]. Mutations in the tumor suppressor genes also contribute to tumor development. For example, mutated p53 loses the ability to sense DNA damage and fails to induce tumor apoptosis and senescence [25]. Approximately, three quarters of cancers express mutated p53 with amino acid substitutions, most of which are concentrated in the DNA-binding domain [26]. These mutations in cancer have paved the way for identifying the novel types of tumor antigens called neoantigens. Containing amino acid sequences derived from mutations, neoantigens are immunologically considered non-self-antigens that can activate high-affinity T-cells. The cytotoxic effect of CTLs induced by mutant p53 peptides is superior to that of wild-type p53 peptides [27]. Although high immunogenicity is an advantage of neoantigens against conventional TAAs, neoantigens have several issues that need to be addressed in clinical practice. The detection and confirmation of immunogenicity of non-shared neoantigens is time- and cost-consuming (tailor-made vaccines). Moreover, neoantigen-based peptides lose their antitumor effects, when a second mutation occurs in the neoantigen sequence. Because neoantigens are derived from frequently mutated tumors, targeting neoantigens has a high risk of antigen loss due to secondary mutations. Since antigen loss is common in tumor immunotherapy [28], it would be useful to select epitopes from non-mutated shared TAAs to switch tumor antigens to another. Because of their immunogenicity to induce antitumor T-cells [13], conventional TAAs can be used as clinical peptide vaccines with high versatility and replaceability when compared to neoantigens.

In classical cancer immunotherapy, CTLs are focused on as the “killer” T-cells against tumors, and HTLs are considered as the “helper” T-cells that support the cytotoxic function of CTLs. Although CTLs are potent in killing tumors, HTLs can directly exert antitumor effects through the production of cytokines such as IFN-γ, TNF-α, granzyme-B, and perforin [29,30,31]. Moreover, several reports have shown that HTLs are more important than CTLs in cancer immunotherapy by the direct tumor killing and education of CTLs or natural killer (NK) cells [32,33,34,35]. Because the support of HTLs to prime CTLs is only applied to high-avidity CTLs [36], high-avidity CTLs can be selectively activated with HTL epitope-combined CTL vaccines, whereas low-avidity CTLs compromise high-avidity CTLs with CTL vaccines alone. Collectively, it would be better to combine HTLs with the CTL epitope to augment antitumor responses with the vaccine. Since HTL and CTL epitopes should be presented by the same type of DCs to fully activate CTLs [36], CTL-included or CTL-linked HTL epitopes, rather than a mix of CTL and HTL epitopes, is appropriate to be presented by the same APCs and achieve the proper help of activated HTLs to CTLs through immune synapse [37]. Although it is possible that tumor-irrelevant HTLs might provide enough support to activate antitumor CTLs [38], it is rational to target tumor-reactive HTLs to obtain a benefit from the direct tumor cytotoxicity of these HTLs [39].

### 2.2. The Factors That Modulate Peptide Immunogenicity

The immunogenicity of tumor antigens depends on a complex series of events, including the isotype variants of antigens, antigen processing/presentation, and the post-translational modification of peptides [40]. Despite the inclusion of HTLs with CTL epitopes, as mentioned above, the length of the peptide itself is an important factor in determining the immune response. The addition of a few amino acids to the N-terminal of the CTL epitope increases its ability to be cross-presented by professional APCs [41]. Because longer peptides require further processing by professional APCs such as DCs, elongated peptides are selectively presented by professional APCs. In contrast, minimal epitopes do not require antigen processing and can be presented by non-professional APCs, which induces T-cell anergy [42]. Because protein-based vaccines and long-peptide vaccines can elicit antipeptide antibodies that have a potential risk of inducing anaphylaxis [43], long peptides should be designed without B-cell epitopes. Although antibodies that can capture antigens facilitates cross-presentation through the Fcγ receptor on DCs [44], the vaccine efficacy and safety should be carefully examined. Self-assembling amino acid sequences are unique amphiphilic peptides, which can form large complexes (20–200 nm), such as nanofibers by hydrophobic clustering and intermolecular hydrogen bond formation [45,46]. This complex can be selectively processed through professional APCs followed by the increased antigen-specific T-cell activation without risking the inclusion of B-cell epitopes [47]. The size of self-assembly nanoparticles can be controlled to 20 nm, irrespective of the peptide composition [48]. The insolubility of the complex can be improved by the insertion of suitable spacer amino acids [49]. The palmitoylation of peptides and their combination with cell-penetrating peptides will further increase the delivery of peptides into APCs through lipid bilayers and electrostatic/hydrophobic interactions, respectively [50,51,52]. Extracellular chaperon proteins such as heat shock protein 70 support the phagocytosis of exogenous antigens by penetrating APCs, followed by antigen processing and presentation [53].

Post-translational modifications are the result of coordinated enzymatic actions and are important in the regulation of cellular metabolism and gene expression. In cancer cells, the phosphorylation of proteins is involved in enhanced signal transduction, which affects the proliferative and metastatic potential of cancer cells. Histone acetylation has also been implicated in the upregulation of tumor-associated proteins, such as p53, c-myc, and survivin. Notably, TCRs can distinguish post-translationally modified peptides, including phosphorylated, acetylated, citrullination, or glycosylated peptides from their relevant wild-type peptides [54,55,56]. In addition, T-cells that recognize these post-translationally modified epitopes escape from negative selection in the thymus [57]. Thus, targeting the post-translational modification of epitopes would be a strategy to selectively target tumors, which aberrantly express post-translationally modified proteins, with a peptide vaccine. Because histone deacetylase inhibitors (HDACis) induce the acetylation of proteins, HDACis could augment the antitumor responses of acetylated p53-reactive T-cells [56]. Accordingly, it would be effective to combine HDACis with a peptide vaccine targeting acetylated TAAs.

Since TCRs can distinguish slight changes in epitopes such as post-translational modifications, TCRs accept the substitution of several amino acids within the epitope peptide. This concept has been widely accepted in mouse melanoma models. Mouse T-cells that respond to the human gp100-derived peptide (KVPRNQDWL) can recognize and kill B16 cells that express mouse gp100 (EGSRNQDWL) [58]. Because human gp100 could induce better antitumor T cells against mouse melanoma than mouse gp100, the substitution of amino acids to increase MHC binding might improve the T-cell responses to peptide vaccines. In addition, human T-cells that react to EGFR-derived peptides can react with homologous peptides derived from human epidermal growth factor receptor 2 (HER-2), HER-3, or c-Met [14], suggesting that peptide vaccines can be applied to tumors which express analogous epitopes to the targeted TAAs. Nevertheless, the reactivity to wild-type, modified, or analogous epitopes should be confirmed in the biological assay.

## 3. Adjuvants and Delivery Timing of Vaccines to Induce Optimal Immune Responses

Because TCR stimulation alone with epitope peptides (signal 1) induces T-cell anergy [59], costimulatory molecules such as CD80/CD28 and CD70/CD27 are required to stimulate T-cells as a second signal [60]. Inflammatory cytokines (e.g., type 1 IFN, IL-2, or IL-12) provide the third signal for the optimal activation of T-cells [61]. Thus, appropriate adjuvants that can activate these signaling pathways are necessary to obtain robust antigen-specific T-cell responses with peptide vaccines.

### 3.1. Conventional Adjuvants

The insufficient activation of an antitumor immune response is responsible for the clinical failures of peptide-based antitumor vaccines [62]. Although incomplete Freund’s adjuvant (IFA) has often been used as an adjuvant for various cancer vaccine clinical trials, the objective response is less than 3% [63]. Because peptide vaccines with IFA trap antigen-specific T-cells at the injection site followed by T-cell dysfunction and deletion [64], it should be of no surprise that a peptide vaccine combined with IFA is not effective in the clinic. As the most widely used vaccine adjuvant, alum mainly activates humoral immunity, instead of cellular immunity [65]. Thus, alum is potent for antibody production against microorganisms, but not for antitumor T-cells, and alternative adjuvants with antitumor peptide vaccines have been explored.

### 3.2. Modern Adjuvants

Because antigen-specific T-cells are significantly activated during infection, mimicking microorganisms with pattern-recognition receptor (PRR) ligands may efficiently activate antitumor T-cells as Sultan et al. elegantly reviewed [66]. Among PRR ligands, toll-like receptor (TLR) ligands have been a great leap forward as adjuvants in cancer immunotherapy. In addition to their ability to induce signals 2 and 3 through APC maturation [67], TLR ligands enhance the cross-presentation of antigens by recruiting MHC from the endosomal recycling compartment [68]. TLR3 (polyinosinic:polycytidylic acid (poly-IC)), TLR4 (lipopolysaccharide and monophosphoryl lipid A (MPL)), TLR5 (flagellin), TLR7 (imiquimod and gardiquimod (GDQ)), TLR7/8 (resiquimod), and TLR9 ligands (CpG) have been considered as immune adjuvants. TLR3 activates professional APCs, followed by the expression of costimulatory molecules (e.g., CD80 and CD86) and the release of proinflammatory cytokines (type 1 IFN and IL-12) [50]. Poly-IC, a ligand for TLR3 and MDA5, is often combined with carboxymethylcellulose and poly-lysine (poly-ICLC) to avoid deterioration [69]. Since poly-ICLC is a potent adjuvant for inducing peptide-reactive CTL responses, a TLR7 ligand—GDQ—induces HTL responses better than other TLR ligands [70]. TLR7 agonists have been used as immune adjuvants in recent clinical trials [71], and the results of its clinical efficacy have been awaited. MPL (Melacine^®^) is a promising adjuvant in preventive cancer vaccines, but its efficacy as a curative vaccine should be further improved [72,73]. It is widely accepted that HTLs assist CTLs through the activation of APCs by the CD40–CD40 L interaction [74]. Accordingly, a CD40 agonist combined with TLR agonists significantly increases T-cell responses to peptide-based vaccinations in murine melanoma models [39,75]. A clinical trial with CD40 agonists, poly-ICLC, and peptide vaccines has been initiated for melanoma patients (NCT04364230). Because poly-IC and/or CD40 agonists induce programmed death-ligand 1 (PD-L1) on DCs, the PD-L1 blockade boosts the effector CTL expansion with poly-IC-combined peptide vaccines, indicating that peptide vaccines and adjuvants have synergistic antitumor effects with ICIs [76,77]. A clinical trial combining a peptide vaccine, poly-ICLC, and a PD-1 antagonist induces antigen-specific CTLs and HTLs [78]; additionally, other clinical trials using this combination therapy are ongoing (NCT02834052 and NCT03362060).

The stimulator of interferon genes (STING) is another intracellular PRR. Tumor DNA drives type I IFN and CTL responses through cyclic GMP-AMP synthase-STING signaling [79]. Wang et al. have shown that a STING ligand augments antitumor CTL responses by a peptide vaccine with poly-IC [80]. In addition to its intratumoral administration in clinical trials (NCT03172936 and NCT03010176), systemically administrable formulations of STING ligands have been developed for metastatic tumors [81]. Liposomes, polymeric nanoparticles, and inorganic materials are combined with STING ligands to minimize systemic toxicity [82].

Granulocyte-macrophage colony-stimulating factor (GM-CSF) is one of the most widely used adjuvants, and the vaccines-containing GM-CSF has been approved by the FDA [83]. The recruitment and activation of APCs are essential for GM-CSF-induced antitumor T-cell responses [84]. Although a clinical trial using GM-CSF has shown prolonged survival in breast cancer [85], the results from other clinical trials are far from satisfactory [86,87,88]. Because GM-CSF might induce myeloid-derived suppressor cells [89,90], which has been observed in clinical trials [88], the application of GM-CSF should be carefully examined. Besides GM-CSF, cytokines such as IFN-α, IL-2, IL-15, IL-21, and IL-12 have been applied in preclinical and clinical studies [91]. The main drawback of directly supplying signal 3 in vivo is that cytokines have a short half-life and should be administered daily in large doses, which may cause severe toxicity and T-cell exhaustion [92]. Various modifications have been made to prolong the half-life, including binding with polyethylene glycol or fusion with tumor- or cytokine-targeting antibodies. The administration of IL-2 complexed with anti-IL-2 antibody (IL-2 complex) proliferates antigen-specific CTLs, overcomes programmed cell death protein 1 inhibition and eradicates tumors in vivo [93]. The prodrug of IL-2 is selectively activated in the tumor microenvironment with tumor-associated enzymes [94]. The direct and safe delivery of signal 3 as adjuvants is feasible with peptide vaccines using these approaches.

Because the downregulation of MHC molecules on tumor cells is associated with poor prognoses in cancer patients [95], an increase in the antigen presentation in tumors can augment T-cell responses by peptide vaccines. Recently, several reports have shown that tumor growth signalings has a negative relationship between antigen and presentation machinery. The EGFR blockade augments the antitumor T-cell responses subsequent to MHC classes I and II upregulation in tumor cells [14]. Similarly, MAP kinase inhibitors increase MHC classes I and class II expression and potentiate the antitumor effects of ICIs [96]. Mouse double minute 2 inhibitors also augment antitumor T-cell responses through the upregulation of MHCs [29]. These results suggest that tumors actively inhibit antitumor T-cells through decreased antigen presentation; additionally, the inhibition of the signaling that downregulates MHCs can be a novel adjuvant strategy to augment T-cell responses by peptide vaccines.

### 3.3. The Timing of Vaccine

The schedule of peptide vaccines in combination with adjuvants, chemotherapy, or radiotherapy is also an important issue. TLR stimulation results in a transient refractory period, so-called TLR tolerance [97]. Koga et al. reported that TLR7 tolerance could be avoided by extending the interval between each vaccine for seven days [98]. The repeated administration of TLR7 agonists should be separated for at least five days to improve the therapeutic efficacy [99]. As the T-cell expansion peaks at approximately seven days followed by the contraction phase, in which some effector T-cells switch to memory T-cells [100], it is rational to prolong the second vaccination for seven days from the priming vaccine.

The elimination of immunosuppressive cells by radiotherapy and chemotherapy might be suitable for administration before vaccination to spare T-cell priming, while immune stimulants that enhance the antitumor effects of vaccines, such as immune checkpoint inhibitors, should be administered with or after the vaccine [101]. Because chemotherapy with paclitaxel and carboplatin results in a marked increase in CTLs on days 12–14 [102], immunotherapy might have robust synergistic effects with chemotherapy when combined at this time. Indeed, peptide vaccines have shown stronger efficacy when administered two weeks after chemotherapy (carboplatin + paclitaxel) in cervical cancer patients [103]. On the other hand, other studies have reported improved antitumor efficacy when low-dose cyclophosphamide is administered concurrently with vaccination [104,105]. Since radiotherapy induces the production of the chemokine ligand CXCL16 from tumors, which recruits CXCR6-expressing CTLs [106], vaccination with/after radiotherapy would be beneficial [107]. Collectively, the timing of the peptide vaccine alone or with other therapies is important for inducing robust antitumor T-cell responses.

### 3.4. The Delivery of Vaccine

The delivery route and vehicles of peptides should be carefully examined to induce suitable T-cell responses. Although the subcutaneous administration is a major route to deliver B cell/antibody-targeted vaccines in the clinic, Sultan et al. have shown that the intravascular administration is significantly better in inducing antigen-specific T-cell responses [46]. The antibody produced by B-cells can be systemically spread via blood stream, whereas T-cells should be migrated to target tumor after education in draining lymph nodes. In addition, tumor-reactive T-cells have generally low affinity than microorganism-reactive T-cells. Thus, it is rational that the systemic administration is required to activate tumor-reactive T-cells by peptide vaccines. In addition to the selective presentation by professional APCs as mentioned above, nanoparticles with self-assembly have several advantages in delivering vaccines. Nanoparticles protect antigens from degradation and prolong antigen presentation [108]. In addition, nanoparticles can provide peptides and adjuvants such as TLR7/8 to the same APCs, which can avoid the unnecessary activation of bystander T-cells [109]. Nanomaterials can activate APCs, which was thoroughly reviewed by Sun and Xia [110]. For example, gold nanoparticles induces IL-1β and IL-18 through inflammasome [111]. The delivery of peptides and adjuvants through liposomes or polymeric nanoparticles is also useful in minimizing the systemic toxicity of STING ligands [82].

## 4. Clinical Evidence of Cancer Peptide Vaccines

Several phase I and phase II clinical trials of cancer peptide vaccines have been conducted since the first discovery of MAGE-A1 [112]. Unfortunately, most phase III studies using peptide vaccines do not show improved survival rates. For example, phases I and II studies initially showed that HER2-derived peptide vaccines, E75 (nelipepimut-S), AE37, and GP2 are safe and appear to show clinical efficacy in breast cancer patients [113,114,115]. However, the phase III clinical trial does not demonstrate improved disease-free survival with nelipeptimut-S and GM-CSF [116]. According to a review by Rosenberg et al. in 2004 [63], the overall response rate of peptide vaccines in clinical trials was 2.9% (*n* = 11/381), which is far from satisfactory.

The efficacy of peptide vaccines partly depends on the types of tumors enrolled in clinical trials. Melanoma contains several immunogenic antigens, leading this tumor to be an ideal immunotherapy target, which has been proven in ICI trials. In melanoma patients, peptide vaccines have shown immune responses along with prolonged survival in some clinical trials [117,118]. A phase III study combining glycoprotein 100 (gp100) peptide vaccine with IL-2 in advanced melanoma showed an improvement in the overall survival, progression-free survival, and median survival rate. The toxicity of the peptide vaccine was consistent with a single IL-2 therapy [119]. However, the phase III trial of a peptide vaccine targeting multiple antigens (gp100, MART-1, and tyrosinase) and GM-CSF against resected high-risk melanoma patients did not improve the relapse-free survival and overall survival, indicating that the immunogenicity of the tumor type is not the only factor for predicting peptide vaccines responses.

Because T-cells expanded ex vivo by peptides could effectively cure cancer in the clinical setting [120], it is plausible that the imitation of the ex vivo environment may improve the activation of T-cells in vivo. As described in the previous sections, the switching of conventional adjuvants such as IFA or GM-CSF to modern adjuvants (e.g., poly-IC) is the key to improving the efficacy of clinical peptide vaccines [39]. The safety of poly-ICLC with peptide vaccines has been demonstrated in several trials. Hilf et al. carried out a phase I trial using patient-tailored vaccines, APVAC1 (shared glioblastoma-associated peptides) and APVAC2 (de novo synthesized patient-specific glioblastoma-associated tumor-mutated peptides) with poly-ICLC against glioblastoma patients and showed significant T-cell activation [121]. Recent early phase trials have suggested the high immunogenicity of peptide vaccines combined with poly-ICLC [122,123], and several clinical trials are ongoing.

The improvement of standard therapy with the addition of immunotherapy is an attractive topic. Several trials have demonstrated the synergistic effect of chemotherapy with a peptide vaccine. In a phase II trial of IMA901 (nine HLA class I-binding TAA peptides with an HLA class II-binding TAA peptide) plus GM-CSF with cyclophosphamide against relapsed or refractory renal cell carcinoma, favorable overall survival was associated with peptide-reactive T-cell responses [124]. In contrast, a phase III study of IMA901 with GM-CSF and sunitinib did not prolong the overall survival in renal cell carcinoma patients [125], indicating that the reduction of immunosuppressive regulatory T-cells by cyclophosphamide may support the activity of peptide-reactive T-cells [124]. A phase I trial combining a vascular EGFR-2 peptide (elpamotide) with gemcitabine was conducted to target patients with advanced pancreatic cancer [126]. The combined therapy was related to the prolonged survival (8.7 months) compared to the gemcitabine monotherapy group (5.7 months). However, a randomized phase II/III clinical trial using a combination of elpamotide and gemcitabine was not effective in patients with advanced pancreatic cancer [127]. Moreover, a phase III study using gemcitabine with GV1001 peptide vaccine did not improve the overall survival in patients with locally advanced or metastatic pancreatic cancer [128], suggesting that cyclophosphamide, but not gemcitabine, would have a synergy with the peptide vaccine.

Clinical studies using peptide vaccines have shown only a few serious adverse events (AEs). The most prevalent AEs include injection site reactions, which are reversible and manageable. Grade 3 hematological AEs are sometimes observed but are mainly due to the progression of cancer [121,129]. Kjeldsen et al. have demonstrated the long-term safety of the indoleamine 2,3-dioxygenase-derived peptide vaccine [130]. They administered the peptide for up to five years, and it was well tolerated without inducing severe AEs. Therefore, peptide-based vaccines are considered safe and well-tolerated therapies.

## 5. Future Perspective

Tailor-made cancer vaccines using neoantigens are a promising cancer immunotherapy to induce high-avidity T-cell responses without developing autoimmune disease [131,132]. Taking advantage of next-generation sequencing and in silico epitope prediction, several trials of personalized cancer vaccines are ongoing. Fang et al. conducted a single-arm, open-label, investigator-initiated clinical trial of a peptide-based neoantigen vaccine (iNeo-Vac-P01) for various solid tumors. They have shown that a vaccine induces a T-cell-mediated immune response targeting tumor neoantigens and promising antitumor efficiency [133]. Ott et al. showed the efficiency of a personal neoantigen vaccine to generate HTLs and CTLs targeting 97 unique neoantigens in patients with melanoma [134]. Subsequently, an open-label, phase Ib clinical trial of a personalized neoantigen-based vaccine (NEO-PV-01) combined with PD-1 blockade was conducted in patients with advanced melanoma, non-small cell lung cancer, or bladder cancer [78]. This study showed the feasibility and safety of combination therapy to induce neoantigen-specific HTLs and CTLs that have a cytotoxic phenotype, epitope spreading activity, and an ability to traffic into the tumor. Although neoantigen is a unique non-self-mimicking target as a peptide vaccine, the high cost and the complicated method of detecting personalized neoantigens remain major hurdles to apply this type of vaccine in the clinic. Since neoantigens are derived from highly mutated cancers, another mutation in the neoantigen sequence may cause antigen loss followed by immune evasion. To apply peptide vaccines to a large population of patients, low prices and ease of synthesis are mandatory. Thus, the identification and selection of highly immunogenic epitopes from common TAAs are desired. In addition, common TAA-based peptide vaccines are easy to switch to other TAA-derived epitopes in the case of antigen loss.

In view of the negative results from the phase III clinical trials with IFA or GM-CSF as an adjuvant, it is necessary to reconsider the adjuvant combined with peptide vaccines. Various clinical trials with effective adjuvants, such as poly-IC, are ongoing. In addition, the strategy to regulate immune escape due to the tumor microenvironment by using combination therapy with other therapeutic agents, such as chemotherapy, surgery, ICIs, radiotherapy, molecular-targeted inhibitors, neutralization of inhibitory cytokines, or other immunomodulators, has been studied in several preclinical models and clinical trials [135,136,137,138,139,140,141,142]. These combination therapies might augment antitumor T-cell responses by inducing epitope spreading, increasing the levels of proinflammatory cytokines, upregulating HLA surface expression and suppressing immune escape systems such as regulatory T-cells, M2 macrophages, myeloid-derived suppressor cells, and negative immune checkpoints [143,144,145,146,147,148]. Various delivery systems of peptides with liposomes, polymeric nanoparticles, carbon nanotubes, micelles, gold nanoparticles, and mesoporous silica nanoparticles have been investigated [149,150,151,152]. The optimization of peptide vaccines by selecting appropriate delivery systems with potent peptides and adjuvants will further enhance the clinical benefit of antitumor immunotherapy.

## 6. Conclusions

In addition to molecule-targeted therapies, immunotherapy is regarded as a rapidly developing field in cancer treatment. Because of the success of checkpoint inhibitors by which the survival of heavily pretreated cancer patients has been increasing [153,154], we have firm evidence that immune cells can fight against cancer. Thus, the vital question that remains to be answered is how to stimulate the appropriate antitumor immunity without eliciting life-threatening autoimmunity. Peptide-based immunotherapy has an advantage against non-specific immune activation due to its tumor-antigen selectivity. To apply this promising approach in clinical practice, several factors which could potentiate the effect of antitumor peptide vaccines should be optimized, such as the identification of epitopes that can induce antitumor HTLs and CTLs, the length of amino acids, the selection of potent adjuvants including PRRs, costimulatory molecule agonists and cytokines, and optimal regimens (dose, schedule, and administration route) [39]. The recent understanding of peptide–adjuvant combinations is useful for establishing efficient cancer vaccines (Figure 1). Therefore, the ideal time has arrived to concentrate our knowledge and experience to open new avenues for developing cancer peptide vaccines that have clinical significance.

## Figures and Tables

**Figure 1 vaccines-10-00070-f001:**
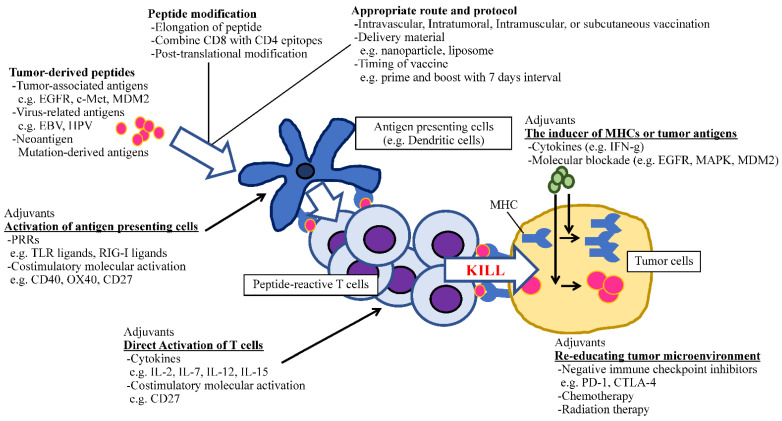
The schematic of optimizing peptide vaccines.

## Data Availability

No new data were created or analyzed in this study. Data sharing is not applicable to this article.

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
