# Peer review of "Antitumor Peptide-Based Vaccine in the Limelight"

_vaccines, 2022, doi:10.3390/vaccines10010070_

Round 1

Reviewer 1 Report

The manuscript written by Kumai et al. reviews the recent therapeutic strategies for developing antitumor peptide-based vaccines.  The manuscript is well written and very informative for groups that do research on that field.

After the reading of the manuscript a few comments have arisen.

First of all, in Section 2 (2. Selecting optimal peptide for cancer peptide vaccine lines 64-75) the listed literature for the prediction algorithms of T-cell epitopes is outdated.  Authors are encouraged to check if there is any progress regarding the prediction algorithms.  If there is such a case it would be useful to add that information to the manuscript.

Also, in the Conclusion Section the first sentence must be rephrased.     

Author Response

Reviewer 1: Thank you for considering our manuscript informative for the researcher in the field of cancer immunology.

>1. First of all, in Section 2 (2. Selecting optimal peptide for cancer peptide vaccine lines 64-75) the listed literature for the prediction algorithms of T-cell epitopes is outdated.  Authors are encouraged to check if there is any progress regarding the prediction algorithms.  If there is such a case it would be useful to add that information to the manuscript.

Answer: We agree with the reviewer. Currently, NetMHCpan-4.1 and NetMHCIIpan-4.0 are considered as reliable methods by combining the data from MHC-binding affinity and mass spectrometry-eluted ligands to predict CD8 and CD4 epitope, respectively (Reynisson et al, 2020). We have included this information in the revised manuscript.

>2. Also, in the Conclusion Section the first sentence must be rephrased.    

Answer: We apologize for the lack of confirmation. We have corrected the indicated sentence in the revised manuscript.

Reviewer 2 Report

Authors highlight the relevance of peptide-based immunotherapy in cancer treatment. The authors discuss various factors that need to be optimized for developing effective anti-tumor peptide based vaccine. Overall, the paper is well written, sound and interesting. Moreover, reckoning with advancements in peptide based cancer vaccines, this is a timely review.

I have a few suggestions to make:

  1. Authors will agree that a good review has pictorial representation of its contents. Therefore, I believe if authors can add one or two figures in the paper, it would make it more comprehensive and presentable.
  2. The authors can also include a section on nanoparticle based delivery of peptide vaccines. Moreover, they can also discuss the adjuvant properties of many delivery vehicles.
  3. Authors can also include addition of self-assembling amino acid sequences to peptide vaccines which can induce self-assembly of the peptide leading to its uptake by APCs and better induction of T-cell immunity.
  4. English language should be checked throughout the manuscript. The first sentence in the 'Conclusion' section should be corrected to begin with uppercase M.

Author Response

Reviewer 2: We thank this reviewer for reviewing and considering our manuscript interesting.

>1. Authors will agree that a good review has pictorial representation of its contents. Therefore, I believe if authors can add one or two figures in the paper, it would make it more comprehensive and presentable.

Answer: Thank you for the valuable suggestion. We agree with the comment, and have added a figure in the revised manuscript.

>2. The authors can also include a section on nanoparticle based delivery of peptide vaccines. Moreover, they can also discuss the adjuvant properties of many delivery vehicles. Authors can also include addition of self-assembling amino acid sequences to peptide vaccines which can induce self-assembly of the peptide leading to its uptake by APCs and better induction of T-cell immunity.

Answer: Thank you for the valuable suggestions. We have added the topic regarding the nanoparticle-based delivery and advantage (selectively processed by professional APCs) of self-assembling peptide in the revised manuscript.

>3. English language should be checked throughout the manuscript. The first sentence in the 'Conclusion' section should be corrected to begin with uppercase M.

Answer: We apologize for the lack of confirmation. The phrase “In addition to” was missed in the “Conclusion” during the English proofreading. We have rechecked the manuscript, and corrected the typos and grammar.